# Lung Fibroblasts from Idiopathic Pulmonary Fibrosis Patients Harbor Short and Unstable Telomeres Leading to Chromosomal Instability

**DOI:** 10.3390/biomedicines10020310

**Published:** 2022-01-28

**Authors:** Radhia M’Kacher, Madeleine Jaillet, Bruno Colicchio, Eirini Vasarmidi, Arnaud Mailleux, Alain Dieterlen, Caroline Kannengiesser, Claire Borie, Noufissa Oudrhiri, Steffen Junker, Philippe Voisin, Eric Jeandidier, Patrice Carde, Michael Fenech, Annelise Bennaceur-Griscelli, Bruno Crestani, Raphael Borie

**Affiliations:** 1Cell Environment DNA Damage R&D, Genopole, F-91058 Evry, France; philippe.voisin@cell-environment.com; 2Inserm U1152, Laboratoire d’Excellence INFLAMEX, Université de Paris, F-75018 Paris, France; madeleine.jaillet@inserm.fr (M.J.); eirini.vasarmidi@inserm.fr (E.V.); arnaud.mailleux@inserm.fr (A.M.); caroline.kannengiesser@aphp.fr (C.K.); bruno.crestani@aphp.fr (B.C.); 3IRIMAS, Institut de Recherche en Informatique, Mathématiques, Automatique et Signal, Université de Haute-Alsace, F-68100 Mulhouse, France; bruno.colicchio@uha.fr (B.C.); alain.dieterlen@uha.fr (A.D.); 4Laboratoire de Génétique, APHP, Hôpital Bichat, F-75018 Paris, France; 5APHP-Service D’Hématologie-Oncohématologie Moléculaire et Cytogénétique Hôpital Paul Brousse Université Paris Saclay, Inserm UMR 935, F-94801 Villejuif, France; claire.borie@aphp.fr (C.B.); noufissa.oudrhiri@aphp.fr (N.O.); annelise.bennaceur@aphp.fr (A.B.-G.); 6Institute of Biomedicine, University of Aarhus, DK-8000 Aarhus, Denmark; sjunker@biomed.au.dk; 7Laboratoire de Génétique, Groupe Hospitalier de la Région de Mulhouse Sud-Alsace, F-68100 Mulhouse, France; jeandidiere@ghrmsa.fr; 8Department of Hematology, Gustave Roussy Cancer Campus, F-94801 Villejuif, France; dr.pcarde@gmail.com; 9School of Pharmacy and Medical Sciences, University of South Australia, Adelaide, SA 5000, Australia; mf.ghf@outlook.com; 10Service de Pneumologie A, Centre de Référence des Maladies Pulmonaires Rares, FHU APOLLO, Hôpital Bichat, APHP, F-75018 Paris, France

**Keywords:** idiopathic pulmonary fibrosis, telomere dysfunction, dicentric chromosome, anaphase bridges, micronuclei, TERT, RTEL1

## Abstract

Idiopathic pulmonary fibrosis (IPF) is associated with several hallmarks of aging including telomere shortening, which can result from germline mutations in telomere related genes (TRGs). Here, we assessed the length and stability of telomeres as well as the integrity of chromosomes in primary lung fibroblasts from 13 IPF patients (including seven patients with pathogenic variants in TRGs) and seven controls. Automatized high-throughput detection of telomeric FISH signals highlighted lower signal intensity in lung fibroblasts from IPF patients, suggesting a telomere length defect in these cells. The increased detection of telomere loss and terminal deletion in IPF cells, particularly in TRG-mutated cells (IPF-TRG), supports the notion that these cells have unstable telomeres. Furthermore, fibroblasts from IPF patients with TRGs mutations exhibited dicentric chromosomes and anaphase bridges. Collectively, our study indicates that fibroblasts from IPF patients exhibit telomere and chromosome instability that likely contribute to the physiopathology.

## 1. Introduction

Telomeres are repeated DNA-sequences (TTAGGG)n associated with a protein complex known as shelterin at the ends of chromosomes. The fundamental role of telomeres is to preserve genome stability [1,2]. Loss of telomere functionality is associated with activation of DNA damage response machinery that can result in chromosome end-to-end fusions and instability [3,4,5]. Human genetic defects that impair telomere length maintenance cause premature age-related diseases, including dyskeratosis congenita, Hoyeraal-Hreidarsson syndrome, and idiopathic pulmonary fibrosis (IPF) [6,7].

IPF is an irreversible interstitial lung disease with no curative treatment and poor prognosis [8]. Germline mutations in telomere related genes (TRG), such as *TERT*, *RTEL1*, *TERC* or *PARN*, are detected in approximately 30% of patients with familial pulmonary fibrosis [9,10,11,12,13]. This observation ascertains for a causal link between impaired telomere length maintenance and predisposition to develop pulmonary fibrosis. Patients with IPF caused by mutations in TRGs can present with additional complications including hematological disorders and hepatic diseases [14,15,16,17]. 

Telomere attrition is associated with genomic instability and predisposes to cancer development [18,19]. Indeed, among 21 acute myeloid leukemia patients with chromosomal rearrangements and aneuploidy *TERT* mutations were found in 18 of those patients, and in cases of aplastic anemia short telomere lengths were associated with numerical and structural chromosome abnormalities which increased the risk of malignancy [20,21]. Moreover, in *Saccharomyces cerevisiae* with telomerase deficiency the rate of spontaneous chromosomal translocations is increased [22]. However, to the best of our knowledge, no data on chromosomal abnormalities have been reported in lung tissue of TRG mutation carriers. An automated approach based on cytogenetic preparations and fluorescence in situ hybridization (FISH) of telomeres and centromeres makes it possible to record telomere aberrations (i.e., telomere loss and terminal deletion) in vast numbers of cells and telomere length variation between samples [23,24].

We hypothesized that chromosomal abnormalities would be more frequent in IPF patients particularly when associated with a TRG mutation, and that telomere length would display a greater variation in addition to the known shortening. Here, we applied this automated approach to primary cultures of lung fibroblasts from IPF patients and from controls in order to assess telomere length and telomere stability.

## 2. Materials and Methods

### 2.1. Lung Fibroblasts

Lung fibroblasts were cultured from lung biopsies obtained from 13 IPF patients undergoing open lung surgery or at the time of lung transplantation, as previously described [15]. All IPF diagnoses were centrally reviewed and confirmed in multidisciplinary discussion [25]. Among the 13 IPF patients, seven IPF patients carried pathogenic variants in a TRG (hereafter noted IPF-TRG), six carried a known *TERT* variant and one had a *RTEL1* pathogenic variant. In the remaining six IPF patients (hereafter noted IPF-noTRG) no variants in known TRGs were detected after next-generation sequencing analysis including *COPA*, *HPS1–4, NKX2.1, SERPINA1, SLC34A2, SLC7A7, SMPD1, TMEM173, ACD, DKC1, GATA2, NAF1, NHP2, NOP10, PARN, RTEL1, TERT, TERC, TINF2, USB1* and *WRAP53* [16,26]. Seven lung biopsies obtained after cancer surgery, were sampled from normal looking lung tissue and confirmed histologically. These biopsies then served as controls. The patient and control features are described in Table 1. The local ethics committee (Comité de Protection des personnes (CPP) Ile de France 1, No. 0811760, 25 March 2010) approved this study. Written informed consent was obtained from all subjects.

Primary lung fibroblasts from controls and IPF patients were cultured as described previously and used at passage 4 [15].

### 2.2. Preparation of Cytogenetic Slides

Cultured fibroblasts were exposed to colcemid (0.1 µg/mL) (Gibco KaryoMAX, Life technologies Europe, Bleiswijk, The Netherlands) for 3 h. After cell harvest and centrifugation for 7 min at 1400 RPM at room temperature (RT), the supernatant was removed and the cells re-suspended in a warm (37 °C) solution of 0.075 M KCl (Merck, Branchburg, NJ, USA) and incubated for 20 min in a 37 °C water bath. For pre-fixation of the cells, approximately five drops of fixative (3:1 ethanol/acetic acid) were added to the cell pellet under continuous agitation and the tubes centrifuged for 7 min at 1400 RPM at room temperature. The supernatant was removed and the cells suspended in fixative solution and centrifuged as before. After two additional rounds of fixation, cells were stored in fixative solution at 4 °C overnight. The metaphases were spread on cold wet slides, dried overnight at room temperature and stored at −20 °C until further use.

### 2.3. Telomere and Centromere Staining

Telomere and centromere staining were performed using Cy-3-labelled PNA probe specific for telomere sequences and FITC-labelled PNA probe specific for centromere sequences, respectively (Eurogentec, Liège, Belgium), as previously described [27].

#### 2.3.1. Telomere Length

Measurement of telomere FISH signal intensity in interphase cells was performed using Metacyte software (MetaSystems, version 3.9.1, Altlussheim, Germany) and TeloScore Software (Cell Environment, Paris, France). The mean fluorescence intensity (FI) of telomeres was automatically quantified in 10,000 nuclei on each slide. The settings for exposure and gain remained constant between captures. The experiments were performed in triplicate. 

#### 2.3.2. Telomere Abnormalities

Analysis of metaphase spreads allowed detection of telomere abnormalities using ChromoScore Software (Cell Environment, Paris, France). The images of metaphases were captured using automated acquisition module Autocapt software (MetaSystems, version 3.9.1) and a ZEISS Plan-Apochromat 63 × 1.40 oil (Zeiss, Oberkochen, Germany) and CoolCube 1 Digital High Resolution CCD Camera (MetaSystems, Altlussheim, Germany) with constant settings for exposure and gain. 

The scored telomere abnormalities were (i) sister telomere loss, likely occurring in G2, and defined as a telomere signal-free end at a single chromatid [4], and (ii) telomere deletion, defined as the loss of two telomere signals on the same chromosome arm (likely resulting from the loss of one telomere in G1/S), an aberration considered to represent double strand breaks, leading to activation of DNA damage response [28]. Automatic scoring of these aberrations was performed using ChromoScore software. One operator validated and excluded the falsely recorded aberrations.

### 2.4. Dicentric Chromosomes, Micronuclei, and Anaphase Bridges Detection

Dicentric chromosomes, i.e., chromosomes with two centromeres, as well as centric ring chromosomes were scored in approximately 50 metaphases from each patient after telomere and centromere staining [29]. The presence of micronuclei and anaphase bridges were analyzed as described previously [29]. Micronucleus assays were performed in the absence of cytochalasin B. 

### 2.5. Statistical Analysis

Data are expressed as mean +/− standard deviation and analyzed using the Wilcoxon–Mann–Whitney rank sum test (comparison of two sub-groups) or the Kruskal–Wallis non-parametric test (comparison of three sub-groups). A *p*-value < 0.05 is considered statistically significant. All statistics analyses were conducted with the R software (ggpubr and plotROC libraries) (R package version 0.4.0 2020).

## 3. Results

### 3.1. Clinical Characteristics of IPF Patients

Patients and controls characteristics are summarized in Table 1. Briefly, patients with IPF tended to be younger than controls, were less frequently smokers, and performed poorly in lung function tests. 

In comparison with IPF-noTRG patients, IPF-TRG patients tended to be more frequently referred to lung transplantation rather than to surgical lung biopsy (6/7 vs. 2/6, *p* = 0.10). Accordingly, lung function test alterations were more severe in IPF-TRG patients than IPF-noTRG patients as assessed by lower forced vital capacity (FVC) (*p* = 0.09) and lower diffusing lung capacity for carbon monoxide (DLCO) (*p* = 0.05).

The *TERT* variants were (c.1630T > C, c.2005C > T, c.2159T > C, c.2287-2 A > G, c.2516C > T, c.3216G > A) and the *RTEL1* variant was (c.1760C > T). All variants are considered pathogenic [16,26]. Four patients had hematological abnormalities (thombocytopenia, *n* = 3, macrocytosis, *n* = 2), but inspection of bone marrow biopsies did not reveal signs of myelodysplasia in any of the patients. Liver function test results were within the normal range in all patients.

### 3.2. Telomere Staining Demonstrate Reduced and Heterogeneous Telomere Lengths in IPF-TRG Fibroblasts

After hybridization with telomere-specific PNA probes, the signals were automatically acquired, and telomere fluorescence intensity was quantified in 10,000 cells from each sample (Figure 1A). The analysis revealed a significant reduction in telomere signal intensity in lung fibroblasts from both categories of IPF patients compared to those in controls (*p* = 2.2 × 10^−16^, Wilcoxon–Mann–Whitney), and from IPF-noTRG lung fibroblasts compared to IPF-TRG lung fibroblasts (*p* = 2.2 × 10^−16^, Wilcoxon–Mann–Whitney). The mean normalized intensity was 0.052 ± 0.029 for IPF-TRG, 0.049 ± 0.017 for IPF-noTRG and 0.059 ± 0.024 controls (Figure 1B). Moreover, we observed a higher heterogeneity of telomere fluorescence intensity in IPF-TRG fibroblasts than in controls (Figure 1C). These observations show that telomere length in lung fibroblasts from IPF patients is significantly reduced as compared to those of control fibroblasts, and telomere length of fibroblasts of IPF-TRG patients is characterized by the presence of higher rate of cells with substantial telomere shortening.

### 3.3. Telomere Staining Shows Telomere Abnormalities in IPF-TRG Fibroblasts 

To further determine the telomere phenotype in IPF cells we analyzed whether lung fibroblasts from patients exhibited signs of telomere instability such as sister telomere loss or terminal deletion (Figure 2A). Analysis of telomeric FISH signals on metaphase spreads revealed a significant increase in the frequency of sister telomere loss in lung fibroblasts from IPF-TRG patients (4.32 ± 1.43 telomere loss/cell) and IPF-noTRG patients (3.29 ± 1.97 telomere loss/cell) as compared to controls (0.84 ± 0.80 telomere loss/cell, *p* = 0.0012 and *p* = 0.014, respectively; Wilcoxon–Mann–Whitney) (Figure 2B). The frequency of sister telomere loss in IPF-TRG cells was numerically higher than in IPF-noTRG cells, although not significantly different (*p* = 0.39). 

Terminal deletions were virtually absent in control cells (0.02 ± 0.03 telomere deletion/cell) while these aberrations, although rare, were significantly increased in IPF-noTRGs lung fibroblasts (0.47 ± 0.52 telomere deletion/cell, *p* = 0.003). Terminal deletions were more frequent in IPF-TRG fibroblasts (2.27 ± 1.43 telomere deletion/cell, *p* = 0.00019 versus controls and *p* = 0.012 versus IPF-noTRG) suggesting severe telomere instability in these cells.

Collectively, these results indicate that telomere instability is more pronounced in lung fibroblasts from IPF patients with known TRG pathogenic variants than in IPF fibroblasts with noTRGs.

### 3.4. Telomere and Centromere Staining Shows Dicentric Chromosomes and Impaired Chromosome Segregation in IPF-TRG Fibroblasts

Since telomere losses can lead to telomere–telomere fusions and generate dicentric chromosomes [5], we asked whether dicentric chromosomes could be detected in lung fibroblasts from IPF patients. To reliably detect dicentric chromosomes, we co-hybridized FISH probes specific for centromere and telomere sequences, respectively (Figure 3A). Dicentric chromosomes were detected in all lung fibroblasts cultures from 7/7 IPF-TRG patients, while such abnormal structures were detected in cells from only one IPF-noTRG patient and never in controls (Figure 3B). The frequency of dicentric chromosomes was significantly increased in IPF-TRG fibroblasts (0.32 ± 0.29 dicentric and ring chromosome/cell) as compared to IPF-noTRG fibroblasts (0.08 ± 0.20 dicentric and ring chromosome/cell) and to controls (0 ± 0 dicentric and ring chromosome/cell, *p* = versus controls and *p* = versus IPF-noTRG).

We analyzed the presence of anaphase bridges and the frequency of micronuclei in IPF fibroblasts and in control cells (Figure 3C). IPF fibroblasts displayed an increased proportion of anaphase bridges (Figure 3D) (0.051 ± 0.038 anaphase bridge/cell in IPF-TRG cells, 0.017 ± 0.017 anaphase bridge/cell in IPF-noTRG vs. 0.002 ± 0.003 anaphase bridge/cell in controls, *p* = 0.0095 and *p* = 0.001, respectively). The fraction of cells with a lagging chromosome or an acentric of chromosome in the form of micronuclei differed also in IPF-TRG patients cells compared to controls (0.084 ± 0.048 micronucleus/cell in IPF-TRG, 0.038 ± 0.038 micronucleus/cell in IPF-noTRG, and 0.002 ± 0.003 micronucleus/cell in controls; *p* = 0.0012 between IPF-TRG and controls).

### 3.5. Aneuploidy in IPF-TRG Fibroblasts Identified by Telomere and Centromere Staining

Defects in chromosome segregation generate aneuploidy in cells during cell division. Given the high frequency of dicentric chromosomes observed in cells with IPF-TRG pathogenic variants, we asked whether these chromosome aberrations could be associated with aneuploidy. Scoring the number of chromosomes on metaphase spreads revealed aneuploidy (either chromosome losses or gains) in 62% of the IPF-nonTRG cells and 60% in the IPF-TRG fibroblasts vs. 11% in the controls (Figure 4). Of note, the presence of large cells with more than 46 chromosomes was detected only in IPF fibroblasts with or without TRG mutations (Figure 4B). These findings were confirmed by analysis of the nuclear area in large number of cells (Figure 4C).

## 4. Discussion

An association between short telomere length, telomere related gene mutations and pulmonary fibrosis was reported in 2005 when a *DKC1* mutation was first described [30]. Our observations on specific pulmonary chromosomal abnormalities in IPF patients and in particular in IPF-TRG patients supports the contribution of telemore instability to the specific IPF pathophysiology. By combining cytogenetic approaches and automated recording of telomere and centromere FISH signals, we have analyzed telomere and chromosomal aberrations in lung fibroblasts from IPF patients with or without TRG mutations. We confirm that lung-derived fibroblasts from IPF patients exhibit shorter telomere length, as assessed by their lower telomere fluorescence intensity than cells from control fibroblasts, and we show for the first time that telomere lengths are more heterogeneous in IPF fibroblasts than in controls. In addition, we also show that TRG mutations in cultured fibroblasts are associated with an increased number of cytogenetic abnormalities in comparison with IPF-noTRG fibroblasts, a novel observation.

We have used a novel approach, which allows simultaneous analysis of telomere dysfunction, chromosomal instability and nuclear morphological changes in thousands of lung-derived fibroblasts in culture. This method could be used to test the effectiveness of treatments targeting a specific mutation or an antifibrotic molecule in vitro. 

TRG mutations are currently found in 30% of the cases of familial pulmonary fibrosis and more rarely in sporadic IPF, although a 10% prevalence of TRG mutations has been reported in sporadic IPF patients prior to lung transplantation [12,13]. TRG mutations are associated with decreased telomerase activity, and causal telomere shortening is detectable in peripheral blood lymphocytes. Reduced telomere length in peripheral blood lymphocytes of IPF patients without TRG mutations have also been reported [31,32], almost all IPF patients having telomere lengths shorter than the median, 25% below the 10th percentile and 10% below the first percentile compared to healthy controls [31]. Moreover, short telomere length has been associated with very poor overall- and post-transplant-survival of IPF patients [33]. Telomere length is usually analyzed in peripheral blood lymphocytes by FLOW-FISH analysis, as PCR may give conflicting results. Interestingly, reduced telomere length has also been reported in situ in lung alveolar type 2 cells and myofibroblasts from IPF patients particularly in *TERT* mutation carriers [34,35,36]. Our automated analysis of nuclear areas and of metaphase spreads combined with FISH enables assessment of telomere stability in lung fibroblasts from IPF patients. We observed that IPF lung fibroblasts, with or without TRG variants, had shorter telomeres compared to control fibroblasts. Interestingly, IPF-TRG lung fibroblasts exhibited the shortest telomeres. These data indicate that telomere shortening in fibroblasts is intrinsic to the fibrosing process, and that this is accelerated, or even more severe, in fibroblasts from patients with a genetic defect in telomere homeostasis. 

In addition to telomere shortening, we found telomere instability in IPF lung fibroblasts, characterized by increased frequency of sister telomere loss and terminal deletion, as compared to control fibroblasts. Noticeably, telomere instability was more frequent in lung fibroblasts from IPF-TRG than in IPF-noTRG, consistent with the roles of *TERT* and *RTEL1* in maintenance of telomere integrity. The cause of short and unstable telomeres in lung fibroblasts from IPF-noTRG patients is not known as yet. One may hypothesize that short and unstable telomeres in lung fibroblasts from IPF-noTRG could result from the local profibrotic milieu that is associated with an increased oxidative stress, activation of the p53 pathway and cellular senescence, a driver of telomere shortening and instability [37,38,39,40]. Another explanation could be an inflammatory condition induced by the cGAS-STING mechanism which is triggered by leakage of DNA from micronuclei into the cytoplasm and/or virus infection of the lungs [41,42].

Apart from telomere abnormalities, we observed several cytogenetic abnormalities, in particular in IPF-TRG fibroblasts. Increased DNA damage assessed by γH2AX staining has already been reported in type 2 cells and myofibroblasts from IPF patients, especially in carriers of *TERT* mutations. Since telomere losses can lead to telomere–telomere fusions and generate dicentric chromosomes [5], we asked whether dicentric chromosomes were present in lung fibroblasts from IPF patients. Indeed, we identified dicentric chromosomes in lung fibroblasts from all the IPF-TRG patients and in fibroblasts from one of the six IPF-noTRG patients, but just a few in controls. Dicentric chromosomes, which likely result from fusion between telomere-free chromosomes, represent an important source of genomic instability [43,44]. Whether genomic instability in this context induces inflammation or promotes fibrosis or cancer need future investigation. 

During mitosis, dicentric chromosomes display increased instability, and in half of the cases formation of anaphase bridges are induced if the two centromeres from the dicentric chromosome are pulled to opposite poles during anaphase [45]. We detected a high frequency of anaphase bridges in cells of patients with IPF-TRG compared to that observed in IPF-noTRG cells or in control cells. In addition, high frequency of micronuclei was scored in cells from IPF patients compared to control cells. In contrast to the analysis of telomere loss, telomere deletion and dicentric chromosomes that require cell culture and metaphase preparations, scoring of anaphase bridges and micronuclei being a morphologic nuclear change can be performed directly on tissues, assuming they are undergoing nuclear division in vivo at the time of sampling [46]. One of the consequences of micronuclei formation is activation of cyclic GMP-AMP synthase-stimulator of interferon genes (cGAS-STING) signaling pathway and induction of inflammatory cytokines, leading to elevated oxidative stress that increases chromosomal instability via vicious cycles of micronuclei, nucleoplasmic bridges and nuclear buds formation [47].

Our work has some limits. Thus, telomere instability and chromosomal abnormalities observed in cultured fibroblasts were not compared to the patient’s circulating lymphocytes or other lung cells. However, the present study is the first to evaluate in detail the cytogenetic abnormalities in lung fibroblasts from sporadic IPF patients and from TRG-variants IPF patients. In addition, fibroblasts were obtained from open lung biopsy or at the time of lung transplantation with various stages of fibrosis that also may have an impact the rate of chromosomal abnormalities compared to early fibrosis in IPF-noTRG. Moreover, control fibroblasts were obtained from normal lung tissue during surgery for lung cancer in smokers, and we cannot rule out the possibility that telomere length and chromosomal abnormalities may have been affected by this pre-malignant environment. Previously, studies on lung tumors and peripheral blood lymphocytes from the same patients demonstrate a prospective correlation between chromosomal instability detected in peripheral blood lymphocytes and lung cancer in smokers suggesting that chromosomal alterations detected in lymphocytes reflect chromosomal damage and tumor initiation in other tissues such as the lung [48,49]. In addition, the role of smoking in the formation of DNA damage and persistent local inflammation has been described [50].

The originality of our approach is a simultaneous analysis of telomere dysfunction, chromosomal instability and nuclear morphological changes of IPF cells. The major advantage of this approach is automation of the process, making it possible to apply as a routine screening assay in future large-scale studies. However, access to lung fibroblasts is rather limited and highly invasive. Therefore, it will be important to analyze the putative presence of these biomarkers also in circulating lymphocytes of IPF patients with adequate control populations.

Screening of telomere instability and nuclear morphological changes, biomarkers of chromosomal instability, over time in IPF patients and their family, underscore the role of telomere defects and genomic instability in the progression of the disease. Certainly, future studies will be necessary to address the following questions: (1) which telomere biomarker or set of biomarkers has the strongest diagnostic value for identifying IPF with TRG mutations? (2) What are the main genetic, environmental and life-style, factors that cause telomere dysfunction and chromosomal instability in IPF patients? (3) To what extent could telomere dysfunction and chromosomal instability in IPF patients be used in the design of new therapeutic strategies? 

## 5. Conclusions

We demonstrate that the high frequency of telomere loss and telomere deletion lead to the formation of dicentric chromosomes, anaphase bridges and morphological changes in IPF fibroblasts with TRG mutations. Our results demonstrate that these IPF characteristics appear to be the best biomarkers to identify IPF with or without TRG mutation. Introduction of these tests in the clinic requires their validation in a large cohort of patients as well as the use of more easily accessible cell types such as circulating lymphocytes. Automation of this process should make its application feasible in large clinical cohorts. The analysis of the impact of telomere instability in DNA damage response in IFP should be investigated in the future. 

## Figures and Tables

**Figure 1 biomedicines-10-00310-f001:**
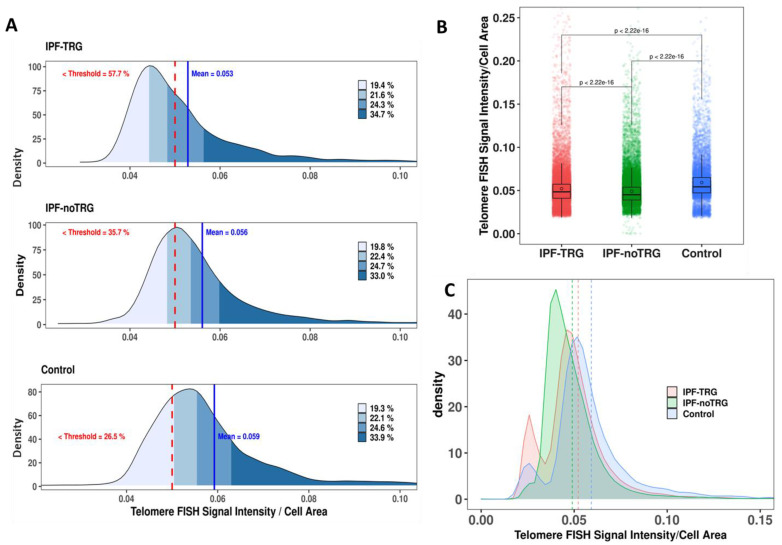
Quantification of FISH signal intensity of telomeres in lung fibroblasts: (**A**) Representative example of quantification of telomere lengths in lung-derived fibroblasts from an IPF-TRG patient, an IPF-noTRG patient and a control. The mean intensity (blue line) is presented, and the frequency of cells with substantial telomere shortening (red dashed line) is presented. The different quartiles of florescence telomere signal intensities are depicted. (**B**) Telomere FISH signals normalized by the area of the nuclei in IPF-TRG fibroblasts, IPF-noTRG fibroblasts and controls. Box plot shows significantly reduced telomere FISH signals in lung fibroblasts from IPF-noTRG patients compared to those of controls, but also in IPF-TRG fibroblasts compared to IPF-noTRG lung fibroblasts. (**C**) Frequency of cells with extreme telomere shortening. The frequency of such fibroblasts of IPF patients and of controls is shown. The highest intercellular variation is observed in fibroblasts of IPF-TRG patients. The mean intensity of telomere signals have been presented (IPF-TRG: pink dotted line; IPR-noTRG: green dotted line; control: blue dotted line).

**Figure 2 biomedicines-10-00310-f002:**
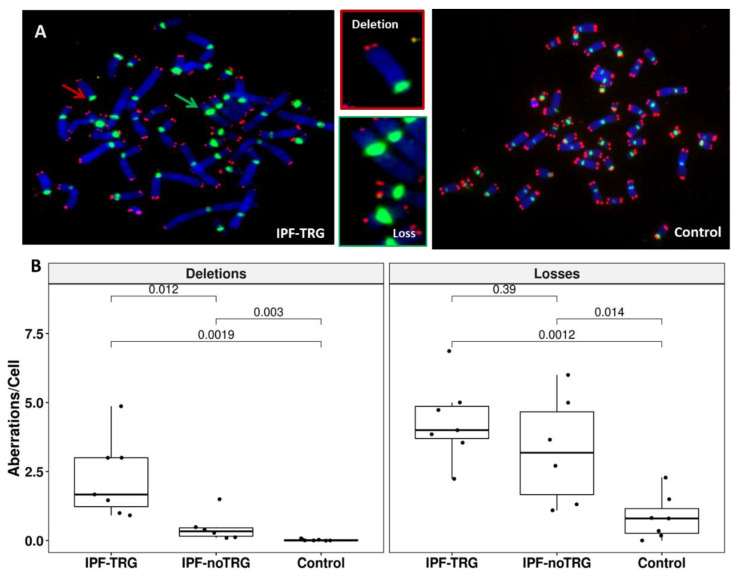
Identification and quantification of telomere aberrations in lung fibroblasts: (**A**) Metaphase spreads from an IPF patient carrying a pathogenic *TERT* variant was stained for telomere (red) and centromere (green) sequences. Metaphase was counterstained with DAPI (blue). The photos depict a high frequency of telomere loss (green arrow) and telomere deletions (red arrow) (63× magnification). (**B**) Box plot demonstrating the significant difference in frequency of telomere deletions between fibroblasts of IPF-TRG patients and fibroblasts of IPF-noTRG patients or controls. Similarly, significant difference was observed between telomere loss in fibroblasts of IPF patients and of controls (*p* = 0.0001), but the difference between telomere loss in cells of IPF-TRG patients and of IPF-noTRG patients was not significant. An average of 50 metaphases were analyzed from each sample.

**Figure 3 biomedicines-10-00310-f003:**
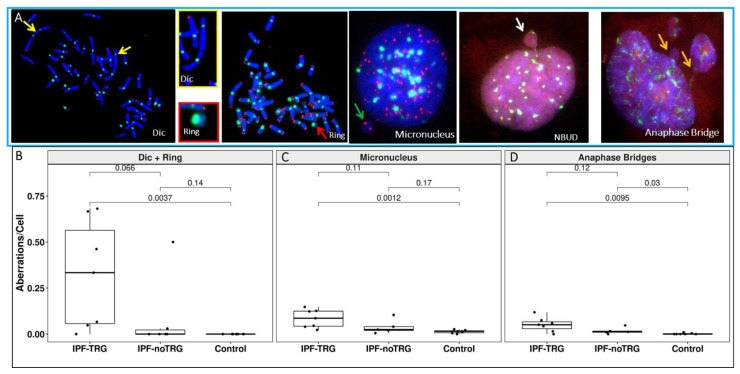
Chromosomal instability in lung fibroblasts of IPF patients. (**A**) Telomere (red) and centromere (green) signals reveal the presence of dicentric chromosomes (yellow arrow), centric rings (red arrow), micronuclei (MN) (green arrow) with only telomere sequences related to the acentric chromosome lagging, nuclear buds (NBUD) (white arrow) and anaphase bridges (orange arrow) formation in IPF-TRG cells (63x magnification). Cells were counterstained with DAPI (blue). (**B**) Significantly higher frequency of dicentric chromosomes and centric rings in lung fibroblasts of IPF patients compared to those of controls as well as IPF-TRG patients and IPF-noTRG patients. (**C**) Frequencies of micronuclei in lung fibroblasts of IPF patients and of controls (**D**) Anaphase bridges as a consequence of formation of dicentric chromosomes were scored. Lung fibroblasts of IPF-TRG patients showed a high rate of anaphase bridge and micronuclei formation.

**Figure 4 biomedicines-10-00310-f004:**
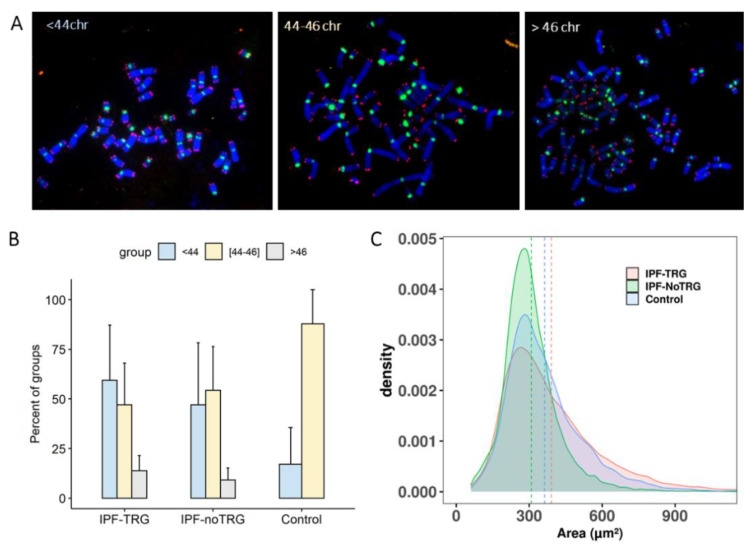
Aneuploidy in lung fibroblasts of IPF patients. (**A**) Three groups of aneuploid metaphases were detected in IPF fibroblasts: (1) metaphases with less than 44 chromosomes; (2) metaphases with 44 to 46 chromosomes; (3) metaphases with more than 46 chromosomes. The scoring of chromosomes were performed after telomere (red) and centromere (green) staining. The metaphase was counterstained with DAPI (blue) (**B**) The frequency of these three groups in IPF fibroblasts compared to controls. IPF fibroblasts were characterized by their small cells (i.e., less than 44 chromosomes) as well as large cells with more than 46 chromosomes. (**C**) Distribution of the area of nuclei in IPF fibroblasts and in controls demonstrating the presence of large nuclei in IPF patients. The mean area of nuclei of IPF-TRG (pink dotted line), IPF-noTRG (green dotted line) and control (blue dotted line) were represented.

**Table 1 biomedicines-10-00310-t001:** Characteristics of patients and controls at the time lung sampling.

	IPF with *TRG* Mutation	IPF without *TRG* Mutation	Controls	*p* *
*n*	7	6	7	
Familial IPF	3	0	0	0.5
Male (%)	4 (57%)	4 (66%)	5 (71%)	1
Ex- or current Smoker	4 (57%)	3 (50%)	7 (100%)	0.05
Age at diagnosis	55.7 (41.7–63.0)	62.7 (56.2–74.2)	64.5 (56.1–75.9)	0.10
Age at lung surgery	58.3 (44.6–65.6)	65.1 (58.1–74.2)	64.5 (56.1–75.9)	0.33
Lung transplantation	6 (88%)	2 (33%)	NA	NA
FVC	56.6 (39.0–75.0)	73.2 (52.0–90.0)	96.1 (80.0–116.0)	0.0007
DLCO	30.6 (15.0–52.0)	47.5 (23.0–59.0)	63.8 (48.0–78.0)	0.002

Data are mean [range] or no. (% of available data). * *p* between the whole group of IPF vs. controls. Forced vital capacity (FVC) and diffusing lung capacity for carbon monoxide (DLCO) are expressed as % predicted value. NA not applicable.

## Data Availability

The data that support the findings of this study are available from the corresponding author upon reasonable request.

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
