# Peer review of "Lung Fibroblasts from Idiopathic Pulmonary Fibrosis Patients Harbor Short and Unstable Telomeres Leading to Chromosomal Instability"

_biomedicines, 2022, doi:10.3390/biomedicines10020310_

Round 1

Reviewer 1 Report

This is an interesting manuscript confirming the known links between IPF and dysfunctional telomeres. The novel findings mostly relate to the evidence of genome instability, that has not been thoroughly characterized in the past.

Although not strictly necessary, but surely able to boost the impact of this manuscript, the additional staining for markers of DNA damage response (DDR) activation such as gH2AX and their telomeric localization would strongly imporve teh originality of the present submission.

Author Response

DNA damage response (DDR) activation such as gH2AX and their telomeric localization

Thank you for this remark, the second article using this cohort of IPF fibroblasts dedicated to DNA damage and the response to genotoxic agent such as irradiation will be submit as soon as possible in this Issue. The following sentence has been included in the conclusion:

“The analysis of the impact of telomere instability in DNA damage response in IFP should be investigated in the future”.

Reviewer 2 Report

Comments to the Author

The author demonstrated telomere and chromosome instability in IPF. The present study has novelty, clinical significance. The novel findings of telomere loss and terminal deletion dicentric chromosomes and anaphase bridges in fibroblast from IPF cases has value of special mention. Please consider the following issue.

Major comments

Please show how to diagnose IPF, including which guidelines it was based on, who made the diagnosis, whether it was facility diagnosis or central diagnosis. Was diagnosis based on MDD diagnosis?

Please show the rate of patients with familial ILD among study patients and comparison of the rate of familial ILD patients between TRG and no-TRG patients.

Author Response

Please show how to diagnose IPF, including which guidelines it was based on, who made the diagnosis, whether it was facility diagnosis or central diagnosis. Was diagnosis based on MDD diagnosis?Please show the rate of patients with familial ILD among study patients and comparison of the rate of familial ILD patients between TRG and no-TRG patients.All patients had a centralised and MDD IPF diagnosis, by experienced ILD specialists (Raphael Borie; Bruno Crestani; Marie Pierre Debray; radiologist; Aurélie Cazes, pathologist) according to the 2018 guidelines for IPF diagnosis.

We suggest including the following sentence in the methods:

“All IPF diagnosis were centrally reviewed and confirmed in multisciplinary discussion (ref).”

Diagnosis of Idiopathic Pulmonary Fibrosis. An Official ATS/ERS/JRS/ALAT Clinical Practice Guideline

Ganesh Raghu , Martine Remy-Jardin , Jeffrey L. Myers , Luca Richeldi , Christopher J. Ryerson , David J. Lederer , Juergen Behr , Vincent Cottin , Sonye K. Danoff , Ferran Morell , Kevin R. Flaherty , Show All... https://doi.org/10.1164/rccm.201807-1255ST       PubMed: 30168753

As suggested by the reviewer the number of familial ILD between TRG (n=3) and no-TRG patients (N=0) is now reported in table 1.